# Spatiotemporal Patterns of Ammonia Nitrogen and Chemical Oxygen Demand in the Huaihe River–Hongze Lake System (Eastern China)

**Jianjun Han** [1,2,3], **Jin Xu** [1,2,3], **Han Chen** [1,2,3], **Pengcheng Xu** [1,2,3] and **Lingling Wang** [1,2,3,*]

[1] State Key Laboratory of Hydrology-Water Resources and Hydraulic Engineering, Hohai University, Nanjing 210098, China
[2] College of Water Conservancy and Hydropower Engineering, Hohai University, Nanjing 210098, China
[3] Key Laboratory of Hydrologic-Cycle and Hydrodynamic-System of Ministry of Water Resources, Hohai University, Nanjing 210098, China
[*] Correspondence: wanglingling@hhu.edu.cn

**Abstract:** Understanding variations in contaminant concentrations and exploring their driving factors are essential for pollution control and water environment improvement. The Huaihe River Basin, as an important region in the eastern region of China, has attracted much attention to its water environment issues in recent years. Therefore, an in-depth analysis of spatiotemporal patterns of water quality parameters was carried out on the Huaihe River–Hongze Lake system, for the period 1998–2018, using the Mann–Kendall test (MKT) and wavelet transforms (WTs). Significant decrease trends of ammonia nitrogen (AN) and chemical oxygen demand (COD) concentrations were detected in the Huaihe River (HR) before 2008 using the MKT. High concentration in the contaminant load was a result of the effect of increased construction and decreased forest on increasing input of pollutants during this period. The results of the WT showed how factors (e.g., streamflow and water temperature), except land use, affect the variations in AN and COD concentrations. The comparison of spatiotemporal patterns of AN and COD between the HR and Hongze Lake (HL) showed their differences in contaminant transport regimes. The contaminants were rapidly transported downstream along the HR with high streamflow during the wet season, while these in the HL were less responsive due to the long residence time of the water body. In addition, rebounds of contaminant concentrations occurred many times at the confluence between the HR and the HL due to strong river–lake interactions, especially in the flood season. These results have implications for future water environment management in the Huaihe River Basin and in similar settings worldwide.

**Keywords:** water quality; river–lake system; wavelet transform; spatiotemporal patterns; driving factors

## 1. Introduction

Lakes play a very important role in freshwater storage systems in surface ecosystems, which are usually connected by many rivers [1,2]. Influenced by the connections and interactions of rivers and lakes, contaminant transport regimes are very complex and usually unpredictable in the watershed with a river–lake system [3,4], which makes it difficult to protect water resources from pollution.

Huaihe River Basin (HRB), located in the eastern coastal area of China, has suffered from severe river and lake pollution in the past decades. Several serious large-scale water pollution incidents in the Huaihe River (HR) have greatly affected the drinking water safety of local residents in the last three decades [5]. In the HRB, contaminant dynamics (source, transport, and transformation) involve nonlinear hydrological processes and diverse human activities, which act on different spatiotemporal scales [6,7]. Effective water environment management requires long-term predictions of water quality parameters and, therefore,

a comprehensive understanding of the spatiotemporal dependency of the contaminant transport process in the HRB.

One of the fundamental questions involved in solving the water environment pollution in the HRB is which factor(s) drive contaminant transport at specific temporal scales. As many studies have demonstrated, water quality constituent concentrations vary at different temporal scales, including event, daily, seasonal and interannual scales [8–11]. At the scale of a water pollution event, many studies have analyzed the correlation between streamflow and water quality parameters through physical-process simulations, and hydrologic and water quality results can be recorded in detail [12,13]. Diamantini et al. [14] found that dissolved oxygen can be seasonally influenced by streamflow and water temperatures, demonstrating that dissolved oxygen and temperature have a long-term significant correlations, highlighting the impact of changes in the streamflow regime on a particular water quality parameter. These studies provide a snapshot of contaminant distribution patterns and their influencing factors at specific timescales. However, the analysis of contaminant transport is strongly affected by the timescale. For example, the influence of streamflow trends on water quality may be neglected if only the annual mean was taken as the statistical data [14,15]. Evidently, it is essential to analyze the relative impact of different factors on water quality parameters at specific temporal scales.

In general, there are two typical categories of model generally adopted in water quality analysis, i.e., physical-process-based models and data-driven statistical models [13]. Physical-process-based models can capture details of contaminant transport and investigate the impact of altered boundary conditions on water quality [12]. However, some problems limit the application of such models, including complex modeling processes, calibration of data scarcity, and demand of computer resources, especially for long and dense water quality series [16]. Data-driven statistical models can provide an effective solution to this condition, and are relatively simple and easy to implement [17–19]. Hence, there are many researchers using statistical methods based on datasets to analyze water quality series.

In recent years, applications of wavelet transforms (WTs), one of the statistical methods, have attracted great attention in revealing the internal structure and temporal variations in hydrological systems [20–23]. With respect to water quality analysis, only a few applications of WT are described in the literature due to the lack of high-frequency measurements. For example, Jiang et al. [24] analyzed periodic variation in water quality in the Lixiahe River, located in the coastal plain of eastern China (2003–2017), and found that the water quality index had multiscale periodic fluctuations of 0.25–5 years. In their study, water quality and water level have a significant positive correlation in the wet season, when rainfall runoff carries a large quantity of non-point-source pollutants. Yan et al. [25] used Morlet wavelet time–frequency correlation analysis in turbidity and water quality parameters of the Haihe River Basin in Hebei Province, China, in 2020. They found that although there was a highly significant correlation between turbidity and dissolved oxygen concentrations, human activities exerted a stronger influences over short timescales and caused local changes in water quality series. Therefore, WT has the potential to describe patterns of contaminant concentrations driven by various factors at multiple timescales.

The present study was undertaken to investigate the spatiotemporal patterns of water quality in the Huaihe River–Hongze Lake system with Mann–Kendall test and wavelet transforms. Ammonia nitrogen (AN) and chemical oxygen demand (COD) were chosen in the present study as research objects, which are the severe pollutants in the HRB system. Over the past twenty years, multiple severe incidents of AN out of standard have threatened local water security [8,26,27]. In addition, streamflow and water temperature time series of many stations and land-use change in the HRB were analyzed. The aims of this paper are to: (a) detect and quantify the trends in AN and COD in the HRB with a complex river–lake system; (b) use materials from 1998 to 2018 to identify the spatiotemporal patterns of AN and COD concentrations at multiple timescales; (c) analyze how streamflow, water temperature and land-use change affect the patterns; and (d) analyze particular patterns of water quality caused by river–lake interaction. The present research intends

to understand variations in contaminant transport and their controlling factors, as well as improve understanding for predicting the water environment in the HRB.

## 2. Material and Methods

### 2.1. Study Region

The HRB has a vast inland plain, which provides a foundation for the development of agriculture and urbanization. This basin is rich in resources, especially coal resources, which promote the development of modern industry. In addition, the HRB is also a transportation hub with three major north-south railway arteries passing through this basin.

The Huaihe River is the third largest river in China, originating from the Tongbai Mountain in Henan Province and flowing through four provinces, from west to east, before reaching the outlet at the Hongze Lake [28]. As shown in Figure 1, the Hongze Lake (HL) is located on the HR, which is the main water source for 20,000 square kilometers of cultivated land and 20 million people downstream. The HR-HL system supports the economic development of the eastern coastal areas of China, and its ecological environment protection has attracted increasing attention. This study focused on the 300 km long sub-reach in the Middle Reaches of the Huaihe River (MRHR), the Hongze Lake, and four main tributaries. The HL flows into the Yangtze River through T1, and into the Yellow Sea through T2. As for the flowing direction of T3 and T4, it is uncertain and controlled by the water level of the HL.

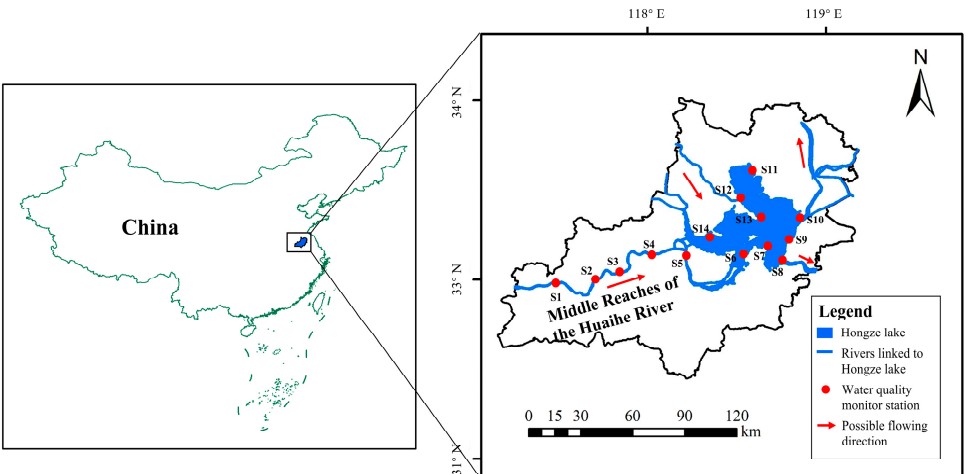

**Figure 1.** Schematic view of the study area.

There are fourteen water quality monitoring stations in this study region. S1 to S6 are located in the mainstream of the HR; S7 to S14 are located around the HL; S6 is in the reach of the HR into the HL; and S8, S10, S12, and S14 are the confluences of the HL and tributaries.

### 2.2. Data

High-frequency and equidistant monitoring data is a prerequisite for the application of WT, the parameters of water quality and temperature were measured weekly or monthly at each station, and flow was measured daily for S1 to S6. The water quality and temperature datasets used in the study were provided by the Huaihe River Water Resources Protection Bulletin and the Huaihe River Water Environment Monitoring Center, which were measured using the national standard water quality detection method. The streamflow dataset was obtained from the Hydrological Bureau of Huaihe River Commission of the Ministry of Water Resources, P. R. C. The time period of the available AN and COD concentration datasets is from 1998 to 2018 for S1 to S4, from 2003 to 2018 for S5 and S6, and from 2004 to 2018 for the other stations (Table 1). As the temperature and streamflow datasets are sufficiently long to match with the water quality data, time series with multiple time steps

can be averaged to analyze the water quality datasets. Because long-term, equidistant time series is a prerequisite for the use of WT analysis [20,29], time series of AN and COD with a week time step were developed at each station. Linear interpolation was utilized for the datasets between 2004 and 2008 at S11 to S14 because they were monthly measured.

**Table 1.** Fourteen monitoring stations in the HR and the HL.

| Station ID | Station Name | Location | Monitoring Period |
|:---:|:---:|:---:|:---:|
| S1 | Lu Taizi | HR | January 1998–December 2018 |
| S2 | Beng Bu | HR | January 1998–December 2018 |
| S3 | Wu Jiadu | HR | January 1998–December 2018 |
| S4 | Lin Huaiguan | HR | January 1998–December 2018 |
| S5 | Xiao Liuxiang | HR | January 2003–December 2018 |
| S6 | Lao Zishan | HR | January 2003–December 2018 |
| S7 | Lin Huai | HL | January 2004–December 2018 |
| S8 | Jiang Ba | HL | January 2004–December 2018 |
| S9 | Gao Liangjian | HL | January 2004–December 2018 |
| S10 | Er Hezha | HL | January 2004–December 2018 |
| S11 | Cheng Zihu | HL | January 2004–December 2018 |
| S12 | Xu Hong | HL | January 2004–December 2018 |
| S13 | Cheng He | HL | January 2004–December 2018 |
| S14 | Li Hewa | HL | January 2004–December 2018 |

Notes: Streamflow datasets are monitored at each station along the HR, while water temperature, AN, and COD datasets are monitored at all stations.

### 2.3. Methods

The following methods were used to analyze the time series of water quality, flow and temperature of the river–lake system: (a) the Mann–Kendall mutation test to detect long-term changes in water quality and streamflow trends over long time scale, (b) the continuous wavelet transform to characterize temporal patterns and to estimate periodicities of the datasets and (c) the wavelet transform coherence analysis to quantify the potential impact of streamflow and temperature on AN and COD concentration.

Mann–Kendall test (MKT [30,31]) is a nonparametric statistical test, which is suitable for testing the trend of non-normal distribution series, and is widely used to detect long-term trends in hydrological and climatic data [30,31]. In this study, the statistic ($S$) of time series $X_t$ with a sample size of $n$ is calculated as:

$$S = \sum_{j=1}^{n-1} \sum_{i=j+1}^{n} \mathrm{sgn}(x_i - x_j) \tag{1}$$

where $x_i$ and $x_j$ are the sequential values (datasets are assumed to be normally distributed with zero mean), and the MKT statistic ($Z$) is given as:

$$Z = \begin{cases} \frac{S-1}{\sqrt{\mathrm{var}(S)}} & S > 0 \\ 0 & S = 0 \\ \frac{S+1}{\sqrt{\mathrm{var}(S)}} & S < 0 \end{cases} \tag{2}$$

where $\mathrm{var}(S)$ is the variance of $S$ at a given significance level $\alpha$. In this test, the original hypothesis would be rejected if $|Z| > Z_{1-\alpha/2}$ ($Z(1 - \alpha/2)$ is the standard value of a normal distribution with probability of $\alpha/2$); thus, there is a significant upward or downward trend for the series $X_t$. The maximum $\alpha$ value used in this study is 10% for $Z > 1.645$, which means 90% confidence level. In this case, the change rate in trends is expressed as Sen's slope $\beta$, which is an efficient estimator of linear trends because it is less sensitive to measurement error [32]. The $\beta$ can be described as follows:

$$\beta = \mathrm{median}\left(\frac{x_i - x_j}{i - j}\right) \quad 1 < i < j < n \tag{3}$$

The continuous wavelet transform (CWT [33,34]) is a common method applied to analyze localized intermittent oscillations in a time series. The CWT often connects two time series together to examine whether regions in time-frequency space with large common power have a consistent phase relationship and, therefore, are suggestive of causality between the time series [22,25,29]. We used Morelet wavelet among many available mother wavelets, which has been proven effective in analyzing hydrological and climatic signals [20]. The statistical significance level of the wavelet power spectrum can be evaluated relative to the null hypotheses that the signal is generated through a stationary process with a given background power spectrum. As many geophysical time series have red noise characteristics that can be modeled very well by a univariate lag-1 autoregressive process, red noise was chosen to analyze the significance level of the wavelet power spectrum [34]. In this study, a confidence level of 95% was used to detect the significant temporal oscillations in the wavelet power spectrum, and a cone of influence (COI) was introduced to represent areas where the spectral amplitude decreases due to signal discontinuity at the edges. Datasets were standardized to eliminate the impact of dimensions before applying the CWT.

Based on the WT, the wavelet transform coherence (WTC) was established to directly measure the correlation between the wavelet power spectra of two nonstationary sequences in time–frequency space [35]. WTC has been widely applied to expose regions with high common power and further reveal information about the phase relationship between two time series in hydro-climate data. The range of WTC values is from 0 to 1, where 0 represents no correlation and 1 represents complete linear correlation [34]. In this study, the WTC was used to investigate the correlation between AN and COD concentration and streamflow and temperature, respectively. In addition, the software used for WT and WTC in this study comes from a MatLab software package in Matlab R2022b, which can be found at https://noc.ac.uk/business/marine-data-products/cross-wavelet-wavelet-coherence-toolbox-matlab (accessed on 13 June 2022).

## 3. Results

This section presents significant trends of AN and COD in the HR and the HL and streamflow in the HR at multiple timescales, as well as spatiotemporal patterns and periodicities of flow, AN and COD concentrations.

### 3.1. Long-Term Trends of Streamflow and Water Quality

Among the six monitoring stations in the HR, the maximum annual average streamflow occurred at Station 5 from 1998 to 2018 (Table 2), which was $886 \pm 1127$ $m^3$ $s^{-1}$, and the maximum recorded peak flow was 8746 $m^3$ $s^{-1}$. In the statistics of all stations, the mean annual AN concentration varied between $0.45 \pm 0.47$ mg $L^{-1}$ (Station 6) and $1.09 \pm 1.24$ mg $L^{-1}$ (Station 3), among which the lowest value occurred at the confluence of the HR and the HL. The mean annual COD concentration varied between $3.78 \pm 0.51$ mg $L^{-1}$ (Station 6) and $4.61 \pm 1.55$ mg $L^{-1}$ (Station 3), and spatial distribution of COD concentration along the HR was consistent with that of AN. In general, the mean concentrations of both AN and COD concentration showed decreased trends along the main channel of the HR, but it increased at S6.

MKT results suggest that annual flows displayed significant increasing trends with 95% confidence level for S1 to S4 (p values are all less than 0.01), while S5 and S6 had no significant trends with 90% confidence level. The AN and COD concentrations of most monitoring stations showed significant downward trends with 95% confidence level, only the COD concentration at S6 had a slight increase of 0.004 mg $L^{-1}$ $y^{-1}$. According to the statistics, the mean concentrations of AN and COD decreased by as much as 41% and 14% at the outlet of the HR, respectively.

**Table 2.** Results of the Mann–Kendall test in the HR for 1998–2018.

| Station ID | Mean Annual Flow (m³ s⁻¹) | Ammonia Nitrogen | | | Chemical Oxygen Demand | | |
|---|---|---|---|---|---|---|---|
| | | Mann–Kendall Test | | | Mann–Kendall Test | | |
| | | Mean (mg L⁻¹) | Z Stat | Sen's Slope (mg L⁻¹ year⁻¹) | Mean (mg L⁻¹) | Z Stat | Sen's Slope (mg L⁻¹ year⁻¹) |
| 1 | 554 ± 612 | 0.76 ± 1.48 | −4.37 * | −0.043 | 4.37 ± 1.35 | −3.54 * | −0.097 |
| 2 | 817 ± 916 | 0.89 ± 1.76 | −4.39 * | −0.064 | 4.04 ± 0.99 | −4.25 * | −0.083 |
| 3 | 778 ± 930 | 1.09 ± 1.24 | −5.35 * | −0.077 | 4.61 ± 1.55 | −3.79 * | −0.127 |
| 4 | 775 ± 933 | 0.92 ± 1.06 | −5.41 * | −0.069 | 4.18 ± 0.98 | −3.21 * | −0.074 |
| 5 | 886 ± 1127 | 0.84 ± 0.97 | −4.27 * | −0.101 | 4.20 ± 1.17 | −2.29 * | −0.106 |
| 6 | 805 ± 1019 | 0.45 ± 0.47 | −4.36 * | −0.049 | 3.78 ± 0.51 | 1.68 ** | 0.004 |

Notes: The results at 95% confidence level are marked with *, and the results at 90% confidence level are marked with **.

As for S7 to S14 in the HL, the mean concentration of AN varied between $0.25 \pm 0.20$ mg L⁻¹ (Station 9) and $0.58 \pm 1.69$ mg L⁻¹ (Station 12) from 2004 to 2018, and COD varied between $2.51 \pm 0.68$ mg L⁻¹ (Station 11) and $4.62 \pm 0.88$ mg L⁻¹ (Station 8), and the water quality was better than stations at the HR (Table 3). MKT results showed that the mean concentration of AN displayed significant downward trends at all stations ($p$ values are all less than 0.01) except S12 and S13, and COD concentration showed significant downward trends with 95% confidence level at S7, S11 and S13 ($p$ values are 0.04, 0.002, and <0.01, respectively), while significant upward trends appeared at S8 ($p < 0.01$), and there were no significant trends in other stations.

**Table 3.** Results of the Mann–Kendall test in the HL for 2004–2018.

| Station ID | Ammonia Nitrogen | | | Chemical Oxygen Demand | | |
|---|---|---|---|---|---|---|
| | Mann–Kendall Test | | | Mann–Kendall Test | | |
| | Mean (mg L⁻¹) | Z Stat | Sen's Slope (mg L⁻¹ y⁻¹) | Mean (mg L⁻¹) | Z Stat | Sen's Slope (×10⁻³ y⁻¹) |
| 7 | 0.26 ± 0.22 | −2.37 * | −0.014 | 3.71 ± 0.62 | −1.97 * | −0.104 |
| 8 | 0.29 ± 0.15 | −3.06 * | −0.016 | 4.62 ± 0.88 | 1.78 ** | 0.024 |
| 9 | 0.25 ± 0.20 | −3.76 * | −0.021 | 3.79 ± 0.53 | 1.23 | 0.003 |
| 10 | 0.34 ± 0.37 | −3.07 * | −0.024 | 3.86 ± 1.69 | −0.98 | −0.001 |
| 11 | 0.26 ± 0.18 | −1.78 ** | −0.008 | 2.51 ± 0.68 | −2.17 * | −0.056 |
| 12 | 0.58 ± 1.69 | −0.69 | −0.011 | 3.89 ± 0.84 | 0.01 | 0.001 |
| 13 | 0.27 ± 0.12 | −1.48 | −0.005 | 4.32 ± 1.31 | −2.78 * | −0.101 |
| 14 | 0.26 ± 0.21 | −2.67 * | −0.019 | 3.41 ± 0.58 | 0.21 | 0.003 |

Notes: The results at 95% confidence level are marked with *, and the results at 90% confidence level are marked with **.

### 3.2. Variability in Streamflow and Annual Water Quality

Figure 2 shows variations in streamflow at S1 station from 1998 to 2018, which can represent the long-term characteristics of the MRHR during this period. Overall, there are differences in annual changes in flow, with typical high-flow years being 1998, 2005, and 2007, and typical low flow years being 2004 and 2014. The upward trends of streamflow over several years can be detected in 2000–2005 and 2015–2018, and downward trends occurred in 2007–2010 and 2011–2014.

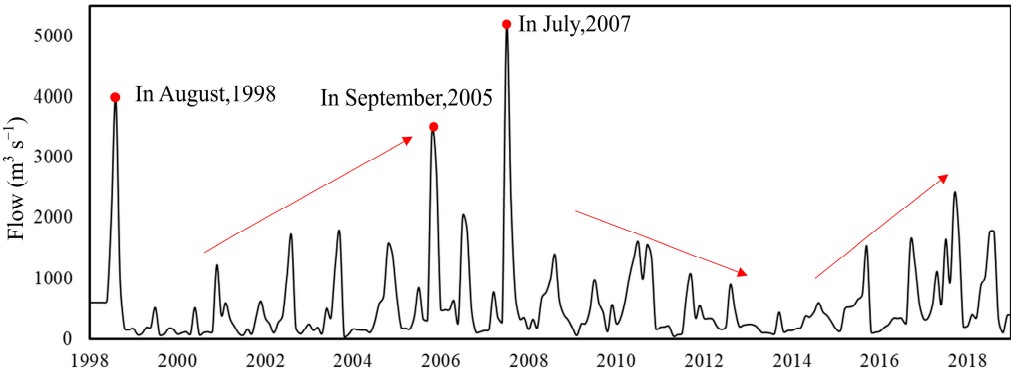

**Figure 2.** Variations in streamflow at S1 for 1998–2018.

Figure 3 shows the annual trends of AN concentration for S1 and S6 in the HR and S12 in the HL, where S6 is the confluence between the HR and the HL. The downward trends were extremely significant at S1 and S6 (*p* values are close to 0) in particular before 2008, and the trends were relatively stable from 2008 to 2018. Despite downward trends also occurring in the HL (Table 3), the annual decrease rate was much lower than stations of the HR. In addition, the AN concentration of most stations (S7 to S14) in the HL had two local peaks between 2004 and 2008. For example, the recorded values in 2012 and 2018 were 0.84 mg L$^{-1}$ and 0.83 mg L$^{-1}$ at S12, respectively, nearly triple that of other years.

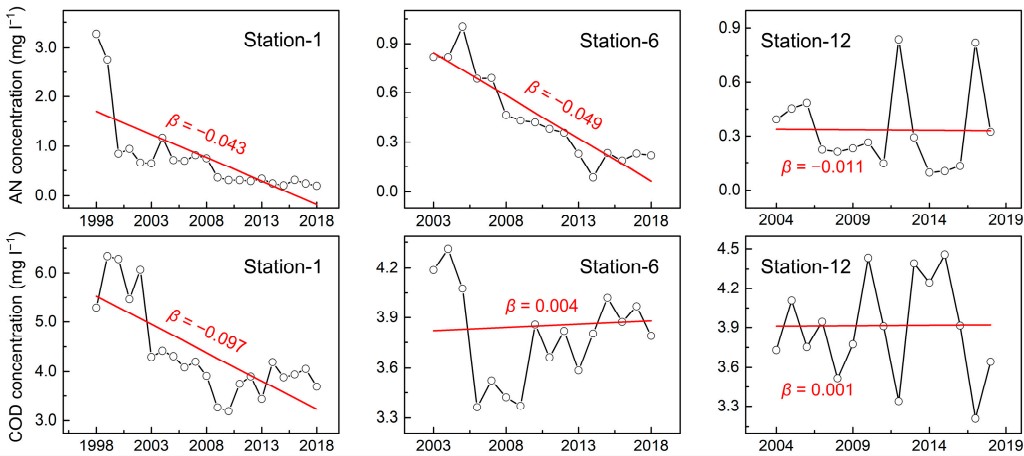

**Figure 3.** Ammonia nitrogen (AN) and chemical oxygen demand (COD) concentrations: annual mean (black line with circle symbols) and long-term trend (red line) for S1 and S6 in the HR and S12 in the HL. The variable *β* is Sen's slope, represented by formula (3).

The variability in annual COD concentration at S1 and S6 was similar to the results of AN, with a demarcation point in 2008 (Figure 3). Nevertheless, the mean annual concentration of COD at S6 rebounded after 2008, which was different to other stations. From 2004 to 2018, the observed COD concentration fluctuated to some extent at S12, which had some abrupt extreme points in this period [8,36].

*3.3. Temporal Patterns and Periodicities*

As indicated in Figure 4, the CWT spectra of streamflow present processes of high magnitude at semiannual and annual timescales in the 2003–2008 and 2015–2018 intervals for all stations. In the ≤0.25-year band, dispersed significant oscillations were detected at S1 to S4, such as the seasonal variations were visible in 2001, 2003–2004, 2010–2013, and 2015–2019 at S1. The wavelet spectrum decreased significantly at S1, S2, S5, and S6 from 2008 to 2015, because this period was the relatively low flow period in the hydrological year of the HR.

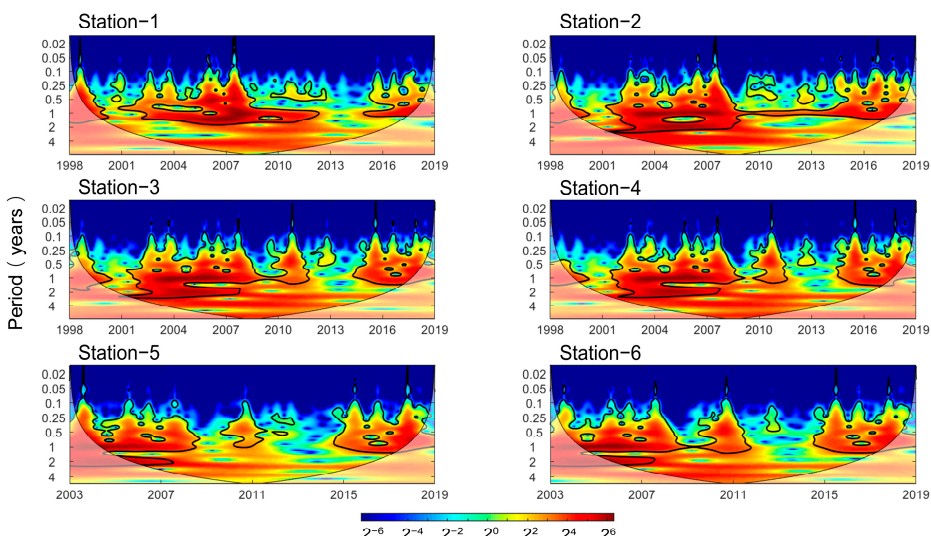

**Figure 4.** CWT spectra for streamflow in the HR, dataset is from 1998 to 2019 for S1 to S4, and from 2003 to 2019 for S5 and S6. The thick black contour encloses regions of greater than 95% confidence for a red noise process. Pale-color areas represent the zone of influence where the edge effects might distort the results.

The CWTs of AN concentration at S1 to S6 (Figure 5) highlight periods of intense AN transport activity at multiple timescales throughout the entire period. Continuous strong oscillations at annual timescales appeared before 2008 at all stations, whereas superimposed oscillations of 0.1–2 years were relatively widespread before 2001 at S1 to S4. Since 2008, the strong oscillation at annual timescales extended up to several years from S3 to S6, especially S4, almost throughout the entire record. The CWTs of the AN concentration at these stations show different patterns of AN transport over the study period the region of high, powerful spectra indicates high magnitude, intense AN transport. Before 2008, the water environment management of the HR had not been improved; thus, the high concentration of AN was often monitored at all stations in the period of 0.1–2 years. In contrast, following water environmental protection policies conducted after 2008, less powerful processes dominated the AN concentration spectra. In addition, there were some discrete AN transport events from S4 to S6 at the time scale of 0.25–0.5 years.

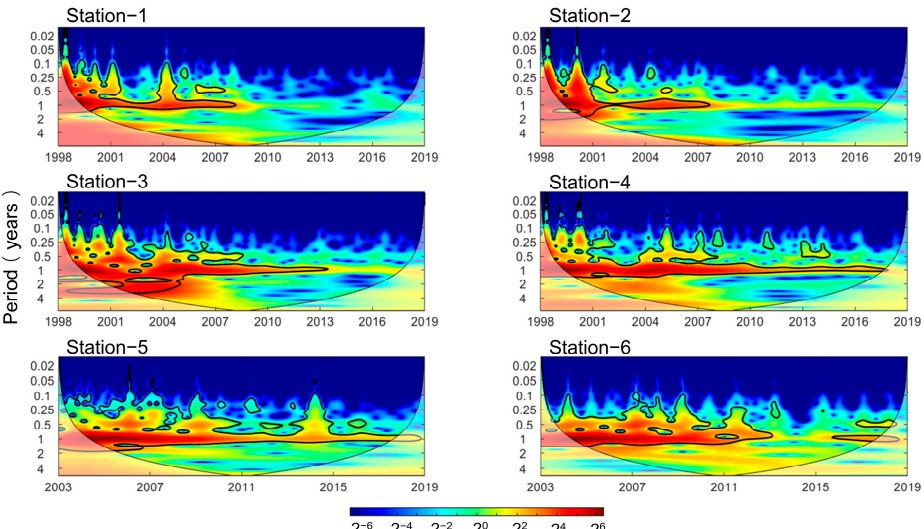

**Figure 5.** CWT spectra for AN concentration in the HR, dataset is from 1998 to 2019 for S1 to S4 and from 2003 to 2019 for S5 and S6.

The wavelet spectra of COD concentration at S1 to S6 (Figure 6) represents periods of COD transport, showing different patterns at multiple time scales. Continuous strong oscillations at the time scale of 0.25–2 years appeared before 2008 at all stations, several pollution events with excessive COD were monitored during this period. In the following period, obvious COD transport in the HR occurred coincidently at specific frequencies and periods. For example, these processes at S3 occurred in 2011, 2014–2016, and 2018. The COD concentration at S6 showed a high-intensity activity almost throughout the whole record, which also led to an upward trend of the annual concentration of COD (Table 2).

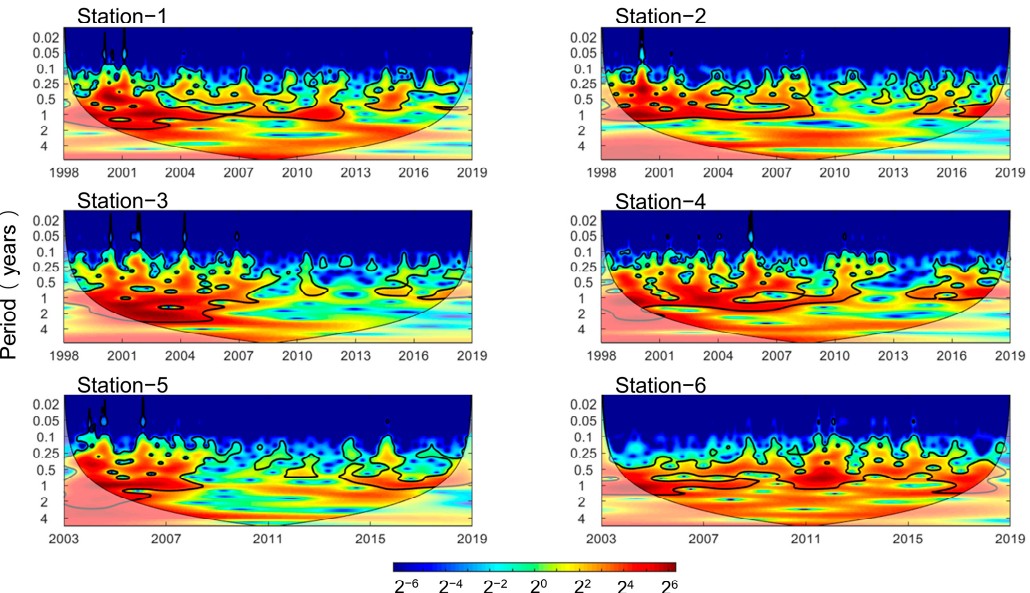

**Figure 6.** CWT spectra for COD concentration in the HR, dataset is from 1998 to 2019 for S1 to S4 and from 2003 to 2019 for S5 and S6.

There were two main contaminant patterns in the HL during the recorded period (Figure 7). At S12 and S14 at the confluence of the HL and tributaries, both AN and COD had high-concentration transport activities at specific times, which was due to the sudden water pollution accidents caused by the HL receiving high-concentration pollutants while receiving water from other tributaries. Therefore, these stations are important nodes for controlling water quality in the HL and should be paid full attention in water quality monitoring. The water quality of other stations in the HL was mainly affected by anthropogenic interventions and water temperature, so their spectra showed contaminant activities at different timescales. We focused our discussion on this pattern of the HL in Section 4 (See Sections 4.2 and 4.3).

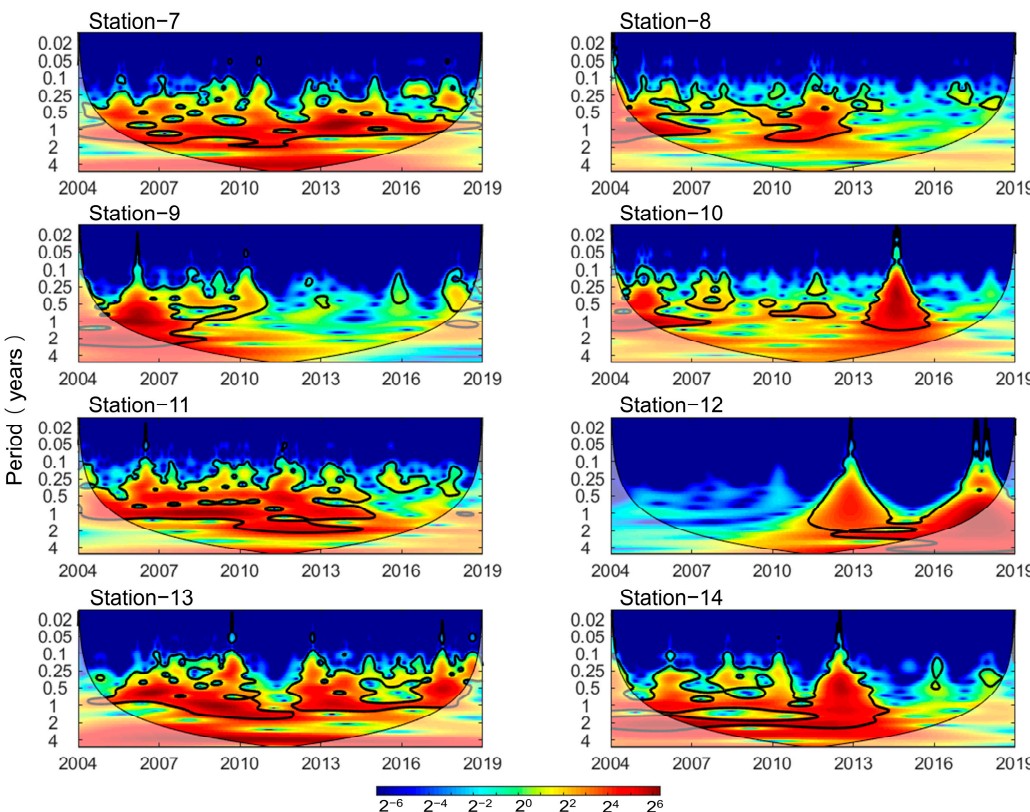

**Figure 7.** CWT spectra for AN concentration in the HL. Dataset is from 2004 to 2019.

## 4. Discussion

### 4.1. The Impacts of Streamflow on Contaminant Transport Regimes

Decreasing trends characterize the annual concentrations of AN and COD of the HR between 1998 and 2018, most of which occurred before 2008. Two of the most important factors influencing the temporal variations in contaminants at the stations are likely to be (a) the emission patterns and routes of river–lake system (anthropic factors), and (b) hydrological events, primarily streamflow events controlling local mutation (natural factors). Over the 20-year period, the main reach of the HR experienced three large floods with peak discharge exceeding 3000 $m^3/s$ in 1998, 2005 and 2007, the maximum peak discharge >7500 $m^3/s$ in 2007 (~20-years flood) since 1954. Contaminant transport becomes slow and accumulates when the streamflow is low in the dry season [14,37]. Then, the first several floods in the wet season will lead to the rapid acceleration of contaminant transport, and the contaminant concentration is more likely to undergo dilution [37]. For example, the monitored AN and COD concentrations at S1 on 7 July 2007 were 0.6 mg $L^{-1}$ and 7.9 mg $L^{-1}$, respectively, and then decreased to 0.2 mg $L^{-1}$ and 4.1 mg $L^{-1}$, respectively, as peak flow increased 2.5 times in three days. Although the temporal patterns of streamflow and contaminant concentration are different, the strong correlation between peak flows and AN concentration can be revealed using wavelet analysis. As seen in Figure 5, low-power oscillations in the 0.5–2 years band of the AN concentration spectra coincided with the relatively dry periods.

There was a lag for the impacts of the rapid increase in peak flow on the concentrations of pollutants. The residual pollutants on the surface will enter the channel under a rainstorm, resulting in the accumulation of a large number of pollutants in the river, and then the peak flow leads to the rapid transport and dilution of pollutants. For example, a small flood with a peak flow of 307 $m^3/s$ occurred at S1 on 24 February 1999, leading to the concentrations of AN and COD suddenly rebounding, and then a rapid decline. A similar event occurred at S2 in this period (Figure 8).

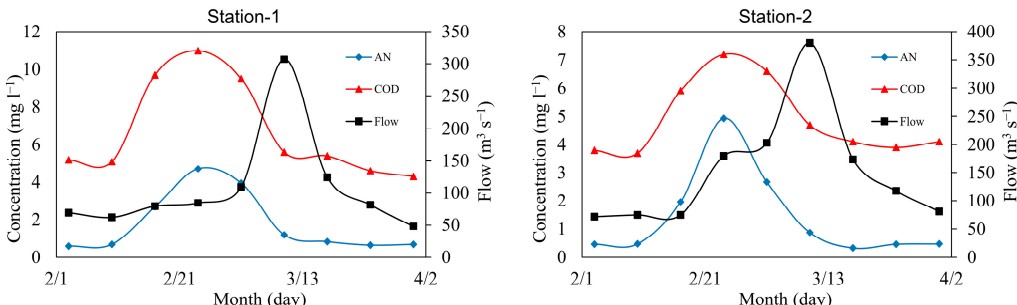

**Figure 8.** The time series of flow, AN and COD concentrations at S1 and S2 in 1999.

The WTC spectra (Figure 9) showed the correlation and noncorrelation characterized the relationship between streamflow and COD concentration at <1-year timescales. The correlation between streamflow and COD concentration was weak over most of the study period, but a significant oscillation at annual time scale took place at S1 around 2011, and the impact of streamflow on AN and COD was transmissible along the HR, which means the high-power oscillations can be found in the same period in S2 to S5. There were also some continuous bands at short time scales (<0.5 years) at all stations, which represent the transport pattern of high-concentration COD in dry seasons, and a large number of non-point-source pollutants entering the river under the effect of rainfall [38].

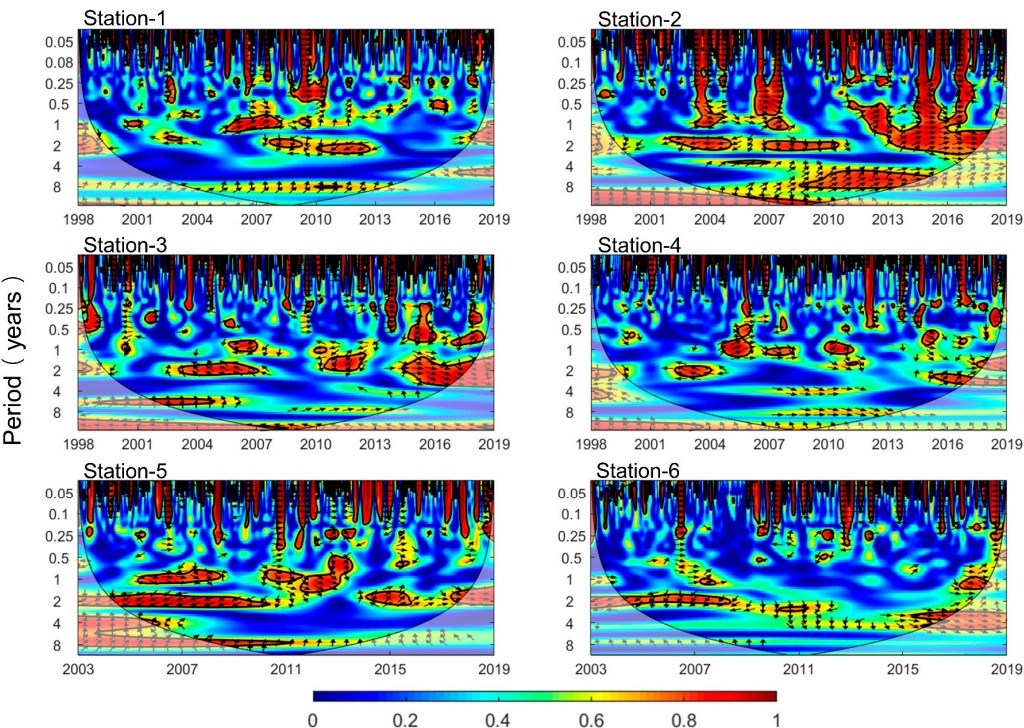

**Figure 9.** WTC between streamflow and COD concentration in the HR, dataset is from 1998 to 2019 for S1 to S4 and from 2003 to 2019 for S5 and S6. The colors indicate the strength of correlation (ranging from 0 to 1) between streamflow and AN concentration. The relative phase relationship is shown as arrows (with in-phase pointing right, anti-phase pointing left, and streamflow leading AN by 90° pointing straight down).

### 4.2. The Impacts of Water Temperature on Water Quality Variability

Water temperature is an important factor to consider when assessing water quality, as it mainly affects contaminant transport rates at low current speeds. The water temperature of the HR mainstream and the HL in the study area shows periodic changes on annual scales during recorded period, and the range of water temperature was 0.11–33.26 °C. The WTC spectra (Figure 10) showed the correlation between temperature and AN concentration at S1 and S6 of the HR and S7 and S8 in the HL. Obviously, the stations in the HR had significant oscillations at the annual scale and almost throughout the entire recorded period, and the period alternated between cold and warm. The coherence is the result of active AN production processes during annual summer months [39,40].

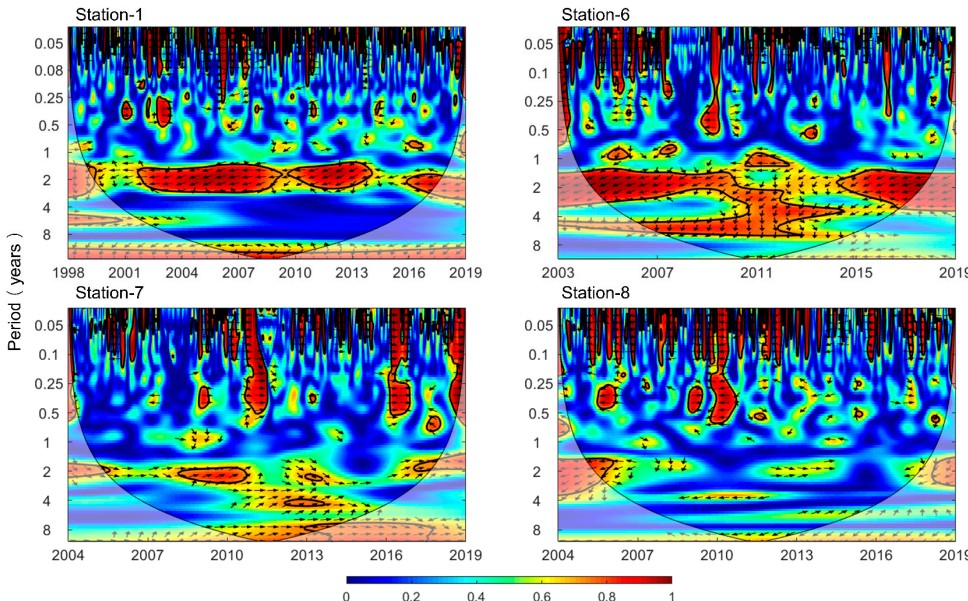

**Figure 10.** WTC between temperature and AN concentration at S1 and S6 to S8, dataset is from 1998 to 2019 for S1, from 2003 to 2019 S6 and from 2004 to 2019 for S7 and S8. Small arrows indicate the relative phase relationship (in-phase, arrows point right; anti-phase, arrows point left).

The HL has a catchment area of 1597 km$^2$ with an average depth of 1.9 m. The lake water only flows slowly when the water is replenished; thus, temperature predominantly influences the water quality of the HL. As a consequence, high-power oscillations appeared in the 0.05–0.25 years bands at S7 and S8, which also confirms the strong response of lake water quality to temperature [41].

In addition, we analyzed the relationship between water temperature and water quality at stations with weak flow in the HL, where water quality was significantly affected by temperature [42]. As the temperature increases, the absorption of nitrogen and oxygen by aquatic organisms is enhanced, and the concentrations of AN and COD decrease. When the temperature exceeds a threshold, the absorption of aquatic organisms will be inhibited. Hence, we found that there were thresholds for the impact of temperature on AN and COD concentrations via quadratic polynomial fitting. The datasets of S7, S9, S11, and S13 were adopted to establish the relationship between water temperature and contaminant concentration during the high-temperature period from June to September, with significant AN and COD activities. As shown in Figure 11, the water temperature thresholds for AN and COD were measured as 18.83 °C and 18.34 °C, respectively. In the range of 16–21 °C, when water temperature exceeded or fell below the threshold, it had an inhibitory effect on the water quality parameters.

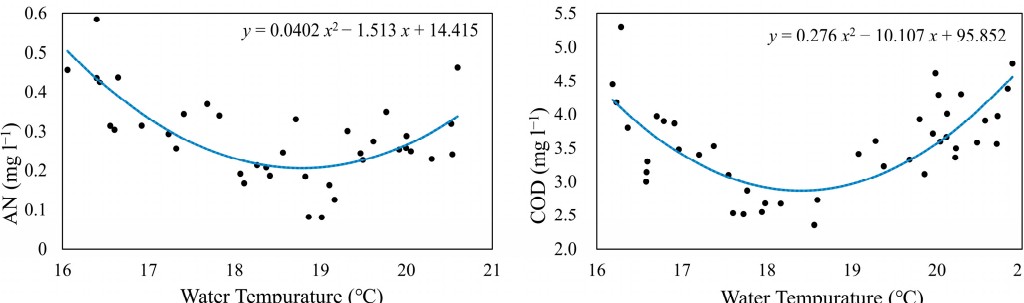

**Figure 11.** Relationship between water temperature and water quality.

### 4.3. The Impacts of Land Use on Water Quality Variability

Water quality is influenced by a combination of natural and anthropogenic factors, the relative influences of which may change over the range of investigated spatiotemporal scales. Land-use changes are often mentioned in research on water environment and ecology, as it directly affects hydrological and water quality changes. In order to assess the impact of land-use change on the water quality distribution patterns, we examined land-use change in the HRB from 1995 to 2015 [43]. In general, as shown in Table 4, the transformation of land-use type can be divided into two stages. Between 1995 and 2000, the development of urbanization of the HRB had just begun, and the rapid growth of population led to a sharp increase in construction, which also brought enormous pressure to the ecosystem. When the grade of urbanization is at a low level, the impacts of land use change on water quality are more significant [44]. During this period, a large amount of forest land was converted into construction land, and the transfer rate was forest land > construction land > water body (Table 4). The functions of forest to degrade pollutants and conserve water resources were greatly weakened due to human activities, resulting in deterioration of water quality, which also explains that the concentrations of AN and COD were high for a long time before 2000 (Figures 3 and 4).

**Table 4.** The area of land-use changes in the HRB for 1995–2015.

| Land-Use Type | 1995–2000 | 2000–2005 | 2005–2010 | 2010–2015 |
|---|---|---|---|---|
| Construction | 1069 | 1598 | 1565 | 1652 |
| Cultivated | −71 | −1724 | −1687 | −1706 |
| Water | 497 | 424 | 42 | 19 |
| Pasture | −29 | −107 | −19 | −21 |
| Forest | −1465 | −110 | 51 | 14 |

Notes: The total area of unused land is small and of little significance, so it is not included. Area is measured in square kilometers ($km^2$).

From 2000 to 2015, the acceleration of urbanization has led to the continuous growth in construction land [8], and the reduction in cultivated land. The transformation of cultivated land types has a direct impact on water quality, and many studies have shown a strong correlation between agricultural activities and water quality parameters, especially AN [44,45]. With the production of agriculture, the use of a large amount of fertilizer increases the concentration of ammonia nitrogen in the soil, which enters river–lake systems under the rainfall. Although the trend of cultivated land transfer to construction land was significant, the impact of this transfer on water quality is not obvious. The phenomenon is similar to current research [43], when cultivated land changes to construction land, and the reduction in agricultural non-point-source pollution can often be offset by urban pollutant discharge. This also explains why the annual variation trends of AN and COD concentrations had remained stable since 2008.

Understanding the relations between land-use changes and water quality response within a watershed is essential to assure sustainable development. The whole watershed land use and their related impacts on water environment cannot be ignored in the HRB's

development strategy. At present, water quality protection is a key water resource management strategy in the HRB, and construction land around the HR-HL system will be carefully evaluated. In short, under artificial governance and policy adjustments, the water quality in this study region will maintain relatively stable and show an improvement trend in the future.

*4.4. Particularity of Water Quality Patterns in a River–Lake System*

The AN and COD patterns of river and lake show spatiotemporal heterogeneity in the river–lake system consisted of the HR and the HL. Their distribution characteristics were affected by factors such as hydrodynamic process, water temperature and anthropogenic interference [4,46]. However, the contaminant transport along the HR was greatly affected by the rainfall and streamflow processes during a rainstorm event. Pollutant concentration recorded at the monitoring stations showed significant oscillations with multiple processes including input by surface pollution source, and dilution due to a large streamflow. The long residence time of the water body in most areas of the HL due to control of the sluices [28], means that the water quality is relatively stable when it is not affected by sudden pollution events caused by human activities. At the same time, this suitable mixing condition of water is conducive to reducing the spatial heterogeneity of AN and COD in the HL.

The confluence of the HL inlet and the HL acts as critical nodes in the river–lake system as it affects flow, water quality, and ecological patterns. Interactions between the HR and HL are undergoing rapid changes due to intensive human activities and ongoing hydrological change [28]. As the largest tributary of the HL, the HR carries a large number of pollutants while supplying water and causes an abrupt change in the spatiotemporal distribution of pollutants at the confluence [8,27]. In the reach of the HR into the HL, hydrodynamic force becomes weak due to water mixing. Negative values of discharge were obtained from July 9th to August 19th in 2017 indicating that the flow direction in the HR reverses and the HL water flows toward the HR. In this condition, the pollutants accumulated at the confluence, resulting in a pollutant concentration recorded at S6 being higher than other stations along the HR (Figure 3), and the effect of backwater will be more significant with the increase in the HL water level.

Under the dual influence of hydrological events and the impoundment of the HL, the increased river–lake interaction and its potential impacts on the river–lake ecosystem raises great concern for the local management authorities. The spatial distributions and mixing patterns of AN and COD had approximately the same features at the confluences between the HL and tributaries, which are mainly manifested as accumulation zones of high-concentration pollution located in the downstream channel of the tributaries. For example, the CWTs of AN time series showed significant oscillations in 2013 at S12, which was related to a major pollution accident in the Xuhong River in November 2012, leading to high-concentration of pollutants into the HL (Figure 7). Hence, future research should focus on the complex interactions between the HR and the HL to better understand the transport of pollutants and solutes at the confluence zone of this river–lake system. In the context of watershed water quality management, it also requires accurate prediction of contaminant transport regimes to understand the spatiotemporal distribution of water quality parameters and take reasonable control measures in the river–lake system.

## 5. Conclusions

The spatiotemporal patterns of AN and COD in a river–lake system are the result of interactions among different drivers at multiple scales. In order to obtain the evolution trend and periodicity of water quality series, the Mann–Kendall test and wavelet analysis were applied in this study. We analyzed the spatiotemporal patterns of AN and COD concentrations in the HR and the HL, and applied wavelet coherence to evaluate the impacts of streamflow, water temperature and land use on them. The analysis revealed AN and COD transport regime in the river–lake system from 1998 to 2018, which reflected the

influence of strong interactions between the HR and the HL on AN and COD patterns at the confluences. The important findings are illustrated as follows:

(1) A significant decrease in the AN and COD concentrations occurred before 2008 in the HR, and the annual trend was relatively stable from 2008 to 2018. In contrast, there were multiple peaks of AN and COD concentrations between 2004 and 2018 in the HL while the overall trend was downward. Our analysis showed the transport of high-concentration contaminants was driven by the decreased forest and increased construction within the catchment.

(2) Wavelet spectral power patterns that demonstrated that streamflow mainly affected the transport process of pollutants in the wet season, and the correlation between peak flows and AN concentration can be revealed using wavelet analysis. In addition, the peak flow generally lagged behind the peak of pollutant concentration.

(3) Water temperature influenced the water quality of river–lake in the long term, and the concentrations of pollutants in the HL were more sensitive to temperature than that in the HL. According to statistics of AN and COD from June to September at S7, S9, S11 and S13, the thresholds for water temperature and water quality in the HL were obtained via polynomial fitting. In summary, the water temperature thresholds for AN and COD were 18.83 °C and 18.34 °C, respectively.

(4) Anomalous AN and COD concentrations (high values and rebounds) were found at the confluence between the HR and the HL due to strong river–lake interactions. The confluences of the HR-HL system act as critical nodes because the water quality at these locations is influenced by tributary flow and the water level of the HL. Our study can improve the understanding of the interactions between this river–lake system and will help develop integrated watershed water environment management strategies for the future.

These findings shed light on the distribution patterns and correlation of AN and COD in the river–lake system in the HRB. In particular, the accumulation of pollutants at the confluence during the flood season needs more attention and control measures. Although the study is based on the HRB, the approach is applicable to evaluate the impact of water quality driving mechanism in other basins.

**Author Contributions:** Methodology, J.H. and J.X.; Software, P.X.; Validation, H.C.; Data curation, H.C.; Writing—original draft, J.H.; Writing—review & editing, J.X. and L.W.; Visualization, P.X.; Project administration, L.W.; Funding acquisition, L.W. All authors have read and agreed to the published version of the manuscript.

**Funding:** This research was funded by the National Key R&D Program of China (2022YFC3202605), the Fundamental Research Funds for the Central Universities (B200204044), and the Research funding of China Three Gorges Corporation (202003251).

**Data Availability Statement:** The data presented in this study are available on request from the corresponding authors. The data are not publicly available due to the continuation of a follow-up study by the authors.

**Conflicts of Interest:** The authors declare no conflict of interest.

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
