# Peer review of "Spatiotemporal Patterns of Ammonia Nitrogen and Chemical Oxygen Demand in the Huaihe River–Hongze Lake System (Eastern China)"

_water, doi:10.3390/w15122157_

Round 1

Reviewer 1 Report

The work is well written and the methodology applied is clear. I believe it would be interesting to increase the resolution of Figures 8 and 9. Furthermore, in the conclusion it would be important to add perspectives from the research work.

Author Response

Point 1: I believe it would be interesting to increase the resolution of Figures 8 and 9.

Response 1: Thank you for the comment. We improved the resolution of Figures 8 and 9 and adjusted the range of the y-axis to make the wavelet power spectra clearer. It can be found in the revised manuscript on page 12 and 13.

Point 2: Furthermore, in the conclusion it would be important to add perspectives from the research work.

Response 2: Thank you for the comment. We improved and supplemented some content in the conclusion, such as providing specific thresholds for the impact of water temperature on water quality.

Reviewer 2 Report

1. In my opinion, to characterize the water quality in the lakes, several parameters must be taken into account. In the present paper, only the concentrations of ammonium nitrate and the chemical oxygen demand were analyzed, whose values do not indicate pollution phenomena.

2. A complex and integrated analysis would require the use of tools such as fuzzy logic or neural networks.

3. The conclusions of the analyzes may be important at the local level, but the research should be extended if a generalization is desired.

Author Response

Point 1:  In my opinion, to characterize the water quality in the lakes, several parameters must be taken into account. In the present paper, only the concentrations of ammonium nitrate and the chemical oxygen demand were analyzed, whose values do not indicate pollution phenomena.

Response 1: Thank you for the comment. Indeedly several parameters such as ammonia nitrogen (AN), DO, COD and phosphorus are important parameters of water quality in rivers and lakes. However, the water quality parameters that are most likely to exceed the standard in the study area are ammonia nitrogen and COD. Large-scale agricultural activities in the Huaihe River Basin have led to multiple water pollution events due to excessive ammonia nitrogen in the past few decades. AN and COD are key indicators concerned by the Huaihe River Water Environment Monitoring Center. As a result, we give an explanation of only condisering the AN and COD, which can be found in line 91-92 on page 2.

Point 2: A complex and integrated analysis would require the use of tools such as fuzzy logic or neural networks.

Response 2: Thank you for your suggestion. It is the best way to use many analysis tools to obtain novel results. However, the present study mainly analyzes the spatiotemporal patterns of water quality, and the Mann-Kendall test and wavelet transforms methods used in this paper have advantages in this regard. At the same time, we take the useful suggestion and we plan to do it in the future work. We are also conducting research on using neural networks for water quality prediction, and we believe that some valuable results will be achieved soon.

Point 3: The conclusions of the analyzes may be important at the local level, but the research should be extended if a generalization is desired.

Response 3: Thank you for the comment. We improved and supplemented some content in the conclusion, such as providing specific thresholds for the impact of water temperature on water quality.

Reviewer 3 Report

This paper is very average. The data are not presented very well in terms of the original datasets so it is difficult to see whether the analysis is correct. The wavelet analysis is ok. The discussion section is not very well done, it is very simple and one-dimensional. The authors have struggled with the English, which I have tried to correct on the scanned pdf. Overall, this paper is adequate but nothing special. 

Author Response

Point 1:  The data are not presented very well in terms of the original datasets so it is difficult to see whether the analysis is correct.

Response 1: Thank you for the comment. The original datasets are too much to list, so we give the most impotant data to verify our analysis, such as station location, data sources, sampling frequency in Section 2.2. We hope the data presented in the revised manuscript could be clear to understand.

Point 2: The discussion section is not very well done, it is very simple and one-dimensional.

Response 2: Thank you for the comment. We have improved the content of the discussion. We divided the research period into two stages around the drivers of land use, and analyzed the impacts of land use on water quality at each stage. The detailed revision can be found in Section 4.3.

Point 3:  The authors have struggled with the English, which I have tried to correct on the scanned pdf.

Response 3: Thank you for providing the scanned pdf, this is very helpful for the improvement of our paper. We have checked the manuscript carefully and improved the language of the manuscript.

Reviewer 4 Report

The spatio-temporal pattern of water quality is affected by many factors. In this paper, the effects of streamflow, water temperature and land use on water quality in Huaihe River Basin were investigated by using Mann-Kendall test and wavelet analysis. The results of this study have certain guiding significance for water environment management in the Huaihe River Basin. However, there are still some deficiencies in the article that need further improvement and English language expression also needs further improvement. Specific suggestions for modification are as follows:

Abstract

In your abstract, the background and purpose of the research are missing. It is recommended to add these two parts so that readers can better understand your research.

Introduction

(1)Line 29import should be important.

(2)Line 41act should be acts.

(3)I think part of the introduction to Huaihe River in the second paragraph can be put into 2.1, or the last paragraph of 1.1 as the reason why you choose Huaihe River Basin as the research object.

(4)Line63Data-driven statistical models should bedata-driven statistical model”

Material and methods

(1) The description of study region is too little. For example, what are the four tributaries respectively? Which of these tributaries go into the lake and which go out? As well as the physical geography and economic situation of the Huaihe River basin should be involved.

(2) It is suggested to add a picture to show the relative position of Huaihe River in China.

(3)The formula should be centered.

(4)You are missing the data processing section, in which you should provide detailed information about how you handled your experimental data, such as what software you used, what method you used to process the data, and mapping.For example, you used the Mann-Kendall test in the results section, you should introduce it in this section

Result

(1)Line196Line206Line208Line214The word significant has a statistical meaning, and if you use the word significant there should be a specific P-value to support your statement.The word also appears in other parts of the article

(2)Line227-229Line273-277:You only need to describe your data objectively without explaining why. This part of the reason should be put in the discussion.

Discussion

Compared with the impact of streamflow and water temperature on water quality, the discussion on the impact of land use on water quality is not deep enough and is relatively simple. This part needs to be further improved, because the impact of land use on water quality has been extensively discussed.

Conclusion

(1) I think your conclusion still does not summarize your research results well and needs further improvement.

Reference

(1)References should be modified strictly in accordance with the format of the journal.For example,    some journal names are lowercase but some journal names are uppercase.

Author Response

Point 1: In your abstract, the background and purpose of the research are missing. It is recommended to add these two parts so that readers can better understand your research.

Response 1: Thank you for the comment. We improve the abstract according to the comment.

Point 2: (1). I think part of the introduction to Huaihe River in the second paragraph can be put into 2.1, or the last paragraph of 1.1 as the reason why you choose Huaihe River Basin as the research object. (2). The description of study region is too little. For example, what are the four tributaries respectively? Which of these tributaries go into the lake and which go out? As well as the physical geography and economic situation of the Huaihe River basin should be involved.

Response 2: Thank you for your kind suggestions. We have adjusted the introduction to Huaihe River in 1.1 to 2.1 and add description of the study region. The description can be found in line104-107 on page 3.

The HRB has a vast inland plain, which provides a foundation for the development of agriculture and urbanization. This basin is rich in resources, especially coal resources, which promote the development of modern industry. In addition, the HRB is also a transportation hub with three major north-south railway arteries passing through this basin.

Point 3: It is suggested to add a picture to show the relative position of Huaihe River in China.

Response 3: Thank you for the comment. We have showed the relative position of Huaihe River in China in Figure 1.

Point 4: The formula should be centered.

Response 4: Thank you for the comment. We have centered all fromulas.

Point 5: You are missing the data processing section, in which you should provide detailed information about how you handled your experimental data, such as what software you used, what method you used to process the data, and mapping.For example, you used the Mann-Kendall test in the results section, you should introduce it in this section.

Response 5: Thank you for the comment. We have supplemented introduction for the software used in wavelet analysis in line190-192. As for the Mann-Kendall test, it is a commonly used statistical analysis method in hydrology, so we used the code for written in python language for data processing.

Point 6: Line196、Line206、Line208、Line214:The word “significant” has a statistical meaning, and if you use the word “significant” there should be a specific P-value to support your statement.The word also appears in other parts of the article.

Response 6: Thank you for your kind suggestion. We have added p-values in the corresponding lines in the paper to support the word 'significant'.

Point 7: Line227-229、Line273-277:You only need to describe your data objectively without explaining why. This part of the reason should be put in the discussion.

Response 7: Thank you for your kind suggestion. We have adjusted these explanations to line443-446.

Point 8: Compared with the impact of streamflow and water temperature on water quality, the discussion on the impact of land use on water quality is not deep enough and is relatively simple. This part needs to be further improved, because the impact of land use on water quality has been extensively discussed.

Response 8: Thank you for the comment. We have improved the content of the discussion. We divided the research period into two stages around the drivers of land use, and analyzed the impacts of land use on water quality at each stage.

Point 9: I think your conclusion still does not summarize your research results well and needs further improvement.

Response 9: Thank you for the comment. We improved and supplemented some content in the conclusion, such as providing specific thresholds for the impact of water temperature on water quality.

Point 10: References should be modified strictly in accordance with the format of the journal. For example, some journal names are lowercase but some journal names are uppercase.

Response 10: Thank you for the comment. We have checked all journal names in the References and standardized the format.

Point 11: (1). Line 29:“import” should be “important”.

                 (2). Line 41:“act” should be “acts”.

                 (3). Line63:“Data-driven statistical models” should be“data-driven statistical model”?

Response 11: Thank you for the comment. We corrected the words and the capitalization.
